# Finding the Gaps: Integrated Serosurveillance and Spatial Clustering of Vaccine Preventable Diseases in Samoa, 2018–2019

**DOI:** 10.3390/tropicalmed11010009

**Published:** 2025-12-28

**Authors:** Selina Ward, Harriet L. S. Lawford, Benn Sartorius, Helen J. Mayfield, Filipina Amosa-Lei Sam, Sarah Louise Sheridan, Robert Thomsen, Satupaitea Viali, Colleen L. Lau

**Affiliations:** 1Faculty of Health, Medicine, and Behavioural Sciences, Center for Clinical Research, University of Queensland, Herston, QLD 4006, Australia; 2School of Medicine, National University of Samoa, Lepapaigalagala Campus, Toomatagi, Samoa; 3National Centre for Immunisation Research and Surveillance—Sydney Children’s Hospitals Network, Westmead, NSW 2145, Australia; 4Samoa Ministry of Health, Moto’otua, Apia, Samoa; 5Oceania University of Medicine, Moto’otua, Apia, Samoa

**Keywords:** integrated serosurveillance, vaccine-preventable diseases, immunity, seroprevalence, Samoa, multiplex bead assays

## Abstract

Introduction: Seroprevalence of antibodies for vaccine-preventable diseases (VPDs), due to vaccination or previous infection, can provide a more accurate estimate of immunity compared to vaccination coverage data alone. This study aimed to examine the seroepidemiology and spatial distribution of VPD seroprevalence in Samoa in 2018 and 2019. Methods: Dried blood spot (DBS) samples were collected from two nationally representative community-based surveys of participants aged ≥5 years from the Surveillance and Monitoring to Eliminate Lymphatic Filariasis and Scabies from Samoa (SaMELFS) project. DBSs were tested using multiplex bead assays (MBAs) to detect antibodies against measles, rubella, diphtheria, and tetanus. Seroprevalence was estimated at the national and primary sampling unit (PSU) levels, and cluster analysis was completed using SaTScan. Results: Overall, 8394 valid MBA results were analysed across 35 PSUs. The highest overall seroprevalence was observed for tetanus (91.0%; 95% CI: 90.2–91.7), followed by diphtheria (83.7%; 95% CI: 82.7–84.7), rubella (79.3%; 95% CI: 78.2–80.3), and measles (45.8%; 95% CI: 44.8–46.9) with substantial heterogeneity across PSUs. Clusters of seronegativity to measles (relative risk [RR]: 1.16, *p* < 0.001) and diphtheria (RR: 1.16, *p* < 0.001) were also identified. Conclusions: These findings demonstrate significant variation in seroprevalence and pockets of low population immunity to multiple VPDs, highlighting the key advantage of an integrated rather than siloed approach. The relatively high seroprevalence to rubella suggests potential community transmission, emphasising the need to strengthen congenital rubella surveillance and improve vaccination coverage. Identifying low immunity to VPDs can provide an early warning to potential outbreak risk and support the Ministry of Health to target public health interventions in higher-risk areas.

## 1. Introduction

Population-level seroprevalence of antibodies (Abs) for vaccine-preventable diseases (VPDs), due to vaccination or previous infection, can provide a more accurate estimate of immunity compared to vaccination coverage data alone [1,2]. Integrated serosurveillance (ISS) is defined by the World Health Organization (WHO) as the “implementation of population-based surveys to collect and analyse samples for simultaneous estimation of seroprevalence to multiple pathogens” [3]. ISS facilitates the systematic assessment of populations experiencing complex and intersecting health disparities, and may serve as an early warning system for outbreak-prone diseases when low seroprevalence is detected [4,5].

To operationalize ISS, multiplex bead assays (MBAs) offer a scalable solution using colour-coded microspheres and fluorescence-based detection to identify Abs against multiple infectious disease antigens simultaneously [6]. One key advantage of MBAs is that only small sample volumes are required; for example, a dried blood spot (DBS) with 10 µL of blood can be used to detect Ab responses for up to 100 different pathogen-specific antigens [7]. This approach has many benefits, including cost-efficiency stemming from a reduced number of analyses required to generate data for multiple pathogens, and logistical streamlining due to integrated data collection [8]. This contrasts with the traditional siloed, single-analyte assay approach with a single-disease focus and facilitates a shift towards multi-pathogen surveillance [9]. In addition, ISS has been used to complement traditional surveillance methods, such as syndromic surveillance, [10], especially in low-resource settings where clinical reporting and/or laboratory facilities may be suboptimal [11]. Such an approach can be cost-effective for governments and policy makers to meet programme targets for disease elimination programmes, such as lymphatic filariasis (LF) and trachoma elimination [12,13].

Seroprevalence estimates of VPDs can be used to monitor the effectiveness of vaccination programmes, providing an accurate comparison of population serological immunity with vaccination coverage based on historical records [2,7,14]. Combining seroprevalence estimates with epidemiological and spatial clustering analysis allows for the targeted identification of geographic clusters of high and low immunity and an understanding of geographic variation in seroprevalence and vaccination coverage [9]. This can help enable targeted public health interventions to specific higher-risk geographic areas [15].

Samoa is a Small Island Developing State (SIDS) in the WHO Western Pacific Region with a population of approximately 205,557 in 2021 [16] residing on the two main islands of Upolu and Savai’i. Samoa is considered a lower-middle income country with high vulnerability to natural disasters and disease outbreaks, and limited health and economic resources [17]. In 1980, Samoa adopted the WHO Expanded Program for Immunization (EPI) to include vaccination for diphtheria, tetanus, pertussis, measles, poliomyelitis, and tuberculosis [18]. In 2003, the measles/rubella (MR) vaccine was introduced. In 2008, this was expanded to include vaccination for diphtheria/tetanus/pertussis, hepatitis B, and *Haemophilus influenzae* type b (DTP-HepB-Hib). In 2009, the measles–mumps–rubella vaccine (MMR) was introduced to replace the MR vaccine [19]. The current standard vaccination schedule in Samoa is included in Appendix A.

The decline in coverage of the first-dose (<1 year age) measles-containing vaccine (MCV) from a reported 99% in 2013 to approximately 30% in 2019 can largely be linked to community mistrust in the safety of vaccination following two vaccine-associated paediatric deaths in July 2018 [20]. This cumulated in the Samoan measles outbreak of September 2019, with over 5700 cases and 83 deaths (mostly in young children) [20]. After the Samoan government declared a state of emergency, mass vaccination efforts were commenced, with an estimated coverage of 90% of the eligible adult population [21]. In 2024, Samoa’s reported vaccination coverage to the WHO was for two doses of MCV at 60% (target ≥95%) and for three-dose coverage of the diphtheria, tetanus, and pertussis-containing vaccine (DTP3) at 85% (target ≥90%) [22]. Despite recent vaccination campaigns, the spatial distribution and population-level immunity gaps that contributed to the measles outbreak remain insufficiently characterised [20].

This study aimed to evaluate the utility of ISS using MBAs for estimating population immunity to VPDs and provide early warning of low immunity. Building on previous work examining seroprevalence to infectious diseases in Samoa in 2018 [23], this study utilises spatial analysis to identify geographic patterns and clustering of seronegativity to VPDs that may warrant further investigation and/or targeted interventions. Using nationally representative MBA data collected from Samoa in both 2018 and 2019, our specific objectives were the following: (i) estimate seroprevalence of four VPDs (measles, rubella diphtheria, and tetanus) at the national and PSU levels; (ii) estimate simultaneous seropositivity to combinations of multiple VPDs (poly-seropositivity); and (iii) identify any clustering of individuals seronegative to VPDs.

## 2. Methods

### 2.1. Ethics Statement

Ethics approvals were granted by the Samoa Ministry of Health (7 February 2019) and The Australian National University Human Research Ethics Committee (protocol 2018/341 approved 11 February 2019) and ratified by The University of Queensland (protocol 2021/HE000895, 16 April 2021). The study was conducted in close collaboration with the Samoa Ministry of Health (MOH), the WHO country office in Samoa, and the Samoa Red Cross. Written informed consent was obtained from adult participants, and verbal assent was obtained from participants aged  <18 years with formal written consent obtained from a parent or guardian [24]. All research activities were performed in accordance with the relevant guidelines and regulations.

### 2.2. Data Sources

This study utilised samples collected from the Surveillance and Monitoring to Eliminate LF and Scabies in Samoa (SaMELFS) project, which was initially designed as an operational research programme to monitor and evaluate the effectiveness of nationwide triple-drug mass drug administration (MDA) on reducing LF prevalence in Samoa [24,25]. Two population-representative community-based cross-sectional cluster surveys were conducted in 2018 (26 September–9 November 2018) prior to MDA distribution (14–26 August 2018), and again in 2019 (8 March–17 May 2019) in 35 primary sampling units (PSUs). Thirty PSUs were selected by systematic random sampling (starting from a random point on a line list of 338 villages in the 2016 national census) with an additional five PSUs purposively selected by the MOH as “suspected hotspots” for LF transmission. Within each PSU, up to 20 households were randomly selected, and household global positioning system (GPS) coordinates were recorded. It is possible that a small subset of households was surveyed in both 2018 and 2019. In addition to household-based recruitment, convenience sampling of 5–9-year-old children was conducted in a central place. For each year, the overall target sample size was 2000 people aged ≥10 years and 2000 children aged 5–9 years, resulting in approximately 57 individuals in each target age group in each of the 35 PSUs [24].

### 2.3. Multiplex Bead Assays

We measured Ab responses to measles (*MeV*), rubella (*RuV*), diphtheria (*Dip*), and tetanus (*Tet*). Laboratory analysis was completed at the US Centers for Disease Control and Prevention (US CDCs). One ear of each DBS (10 µL) was eluted into a 96-well plate and then diluted to a final concentration of 1:400 [26]. Samples were read using a Bio-Plex 200 instrument (Bio-Rad, Hercules, CA, USA). In brief, the fluorescent signal emitted was expressed as the median fluorescence intensity minus background (MFI-bg). MFI-bg cutoffs indicative of immunoprotection were calculated using dilution series of WHO international reference sera to determine thresholds to ≥0.01 IU/mL, ≥0.1 IU/mL, and ≥1 IU/mL Ab levels, where applicable, as shown in Table 1. Information regarding seropositivity cut-off values can be found in Appendix A. All VPDs were examined for ≥0.01 IU/mL l Ab levels, and tetanus and diphtheria were also examined for ≥0.1 IU/mL Ab levels. Only tetanus was examined for ≥1 IU/mL Ab levels.

For all VPDs, ≥0.01 IU/mL Ab levels suggest previous vaccination or past infection but may not be sufficient for full disease protection [27]. Specifically, ≥0.01 IU/mL Ab levels for tetanus and diphtheria are considered to provide minimum protection against infection, whereas a level of ≥0.1 IU/mL corresponds to higher protection against symptomatic infection of the respective diseases [28]. Similarly, for measles and rubella, ≥0.01 IU/mL shows good correlation with protection against symptomatic disease; however, cases of secondary vaccine failure may still occur [29]. For tetanus and diphtheria, ≥0.1 IU/mL Ab levels also suggest previous vaccination or past exposure but are generally considered sufficient for disease protection [30]. For tetanus, ≥1 IU/mL Ab levels are considered highly protective and are associated with long-term immunity [30,31].

### 2.4. Statistical Analysis

Overall seroprevalence: Data cleaning and analysis was performed in Stata (StataCorp, Version 17.0, College Station, TX, USA). Seroprevalence estimates from the 2018 and 2019 surveys were similar (Appendix A) and, given the short interval and the absence of major VPD events between surveys, data from both years were combined to calculate overall seroprevalence estimates and 95% confidence intervals (CIs). For national-level estimates, weighting for selection probability and survey design were performed using the “svyset” command with PSU as the unit of clustering, and standardised for age and sex [24] using the 2016 Samoa Census [32]. Sub-analysis of immunity to multiple VPDs was conducted using the “venndiag” package in Stata. Seropositivity profiles for individuals were determined, with those seropositive to more than one VPD considered poly-seropositive.

Seroprevalence by age and gender: Seroprevalence estimates and 95% CIs were calculated with weighting for selection probability and survey design, with PSU as the unit of clustering. Age group (5–9; 10–19; 20–29; 30–39; 40–49; 50–59; 60–69; 70+ years) estimates were standardised for sex.

Seroprevalence by PSU: At the PSU level, seroprevalence estimates and 95% CIs were adjusted for selection probability and study design, and standardised for age and sex using the 2016 Samoa Census [32]. In brief, specific to children 5–9 years, we adjusted for selection probability based on number of children surveyed (both convenience and household surveys) and the number expected in that PSU. Seroprevalence estimates were mapped using ArcGIS Pro (version 3.1.0).

Spatial clustering: Spatial clustering was performed for ≥0.01 IU/mL protection levels for all VPDs. Spatial cluster analysis was conducted at household location level for participants with GPS locations, using SaTScan™ (Version 10.2.5, Harvard Medical School) [33]. We employed the Bernoulli model with round windows to identify significant (*p* < 0.05) clusters of non-immunity (seronegatives) and calculated relative risk based on the observed cases compared to expected cases. The maximum spatial cluster size was set at 50% of the population at risk, with the number of Monte Carlo replications set to 999. Spatial clusters were mapped using ArcGIS Pro (version 3.1.0).

Definitions of sex and gender: For standardisation of estimates to the population sex distribution, data from the national census was used. When conducting fieldwork surveys, the questionnaire asked specifically “Sex—choose (male, female)”. Therefore, in this analysis, we have used the term “sex” to refer to how the data was standardised and the term “gender” when referring to the data analysis and results.

## 3. Results

### 3.1. Study Population Demographics

In total, 8849 participants were recruited (3940 in 2018 and 4909 in 2019) with a mean age of 19.9 years (18.9 years in 2018; 20.1 years in 2019). Overall, 48.4% of participants were male. Household coordinates were not collected for the 3203 children aged 5–9 years (1542 in 2018 and 1661 in 2019) who were recruited by convenience sampling. Comparatively, 6126 participants (2878 in 2018 and 3248 in 2019) were recruited from households, of which 5216 (85.1%) had GPS data recorded. Valid MBA results were available for 8394 (94.85%) participants (3851 in 2018 and 4543 in 2019).

### 3.2. Overall Seroprevalence and Polyseropositivity

Population-level seroprevalence estimates for minimum protection (≥0.01 IU/mL) for 2018 and 2019 combined were 45.8% (95% CI: 44.8–46.9) for measles, 79.3% (95% CI: 78.2–80.3) for rubella, 83.7% (95% CI: 82.7–84.7) for diphtheria, and 91.0% (95% CI: 90.2–91.7) for tetanus. Seroprevalence to ≥0.1 IU/mL and ≥1 IU/mL levels of protection for diphtheria and tetanus are shown in Table 1. We found that 3562 (38%) individuals were seropositive to all VPDs. Overall, 102 individuals (mean age and standard deviation 11.4 ± 7.6 years; 50.9% female) were seronegative to all the included VPDs (further information can be found in Appendix A). Polyseropositivity to combinations of VPDs are shown in Figure 1.

Seroprevalence by age and gender: Seroprevalence estimates with 95% CI by gender and age groups are shown in Appendix A. Seroprevalence trends by age group are shown in Figure 2. The greatest disparity in seroprevalence was seen between measles and rubella, particularly among 20–29-year-olds (measles 17.4% [95% CI: 14.39–21.09] and rubella 81.05% [95% CI: 76.04–85.22]).

### 3.3. Seroprevalence by PSU

Seroprevalence estimates at the PSU level are shown in Table 2 and Figure 3, respectively, with 95% CIs presented in Appendix A. In Table 2, high heterogeneity in seroprevalence estimates to VPDs can be seen across PSUs, including geographically proximate areas. For measles, the average seroprevalence across all PSUs was 43.9% (standard deviation (SD): 8.2) with a range between 31.0% and 55.4%. Average PSU seroprevalence to rubella was 78.9%; SD: 6.3), with PSU seroprevalence ranging between 66.5% and 88.8%. For diphtheria (≥0.01 IU/mL), average PSU seroprevalence was 84.3% (SD: 5.7), with PSU seroprevalence ranging from 73.8% to 94.1%. The average PSU seroprevalence to tetanus (≥0.01 IU/mL) was 91.6% (SD: 3.5), ranging from 86.1% to 97.5%, with 28 PSUs showing seroprevalence over 90%.

### 3.4. Spatial Clustering

Two statistically significant clusters were identified. The largest cluster was observed for measles seronegativity (population: 1911; observed/expected: 1.10; relative risk [RR]: 1.16, *p* < 0.001) followed by diphtheria seronegativity (population: 847; observed/expected: 1.43; RR: 1.16, *p* < 0.001). Statistically significant clustering was not found for seronegativity to rubella or tetanus. A visual representation of the overlap of these clusters is shown in Figure 4 with smaller non-significant clusters also shown.

## 4. Discussion

Our study demonstrates the utility of ISS to estimate population immunity to VPDs and how spatial analysis has the potential to provide early warning on geographic areas at higher risk of outbreaks and immunity gaps. Key findings include marked discrepancies between seroprevalence to measles and rubella despite the use of combination vaccines, the identification of individuals seronegative to all VPDs (measles, rubella, diphtheria, and tetanus), and overlapping geographic clusters of measles and diphtheria seronegativity. These results highlight the added value of examining multiple VPDs simultaneously to better understand immunity gaps and inform targeted interventions.

The use of seroepidemiology for monitoring population immunity to VPDs is a rapidly emerging field [1,34,35] and testing for multiple VPDs simultaneously can reveal patterns that may be overlooked when undertaking a siloed approach. Our analysis of seropositivity to different combinations of VPDs identified a notable discrepancy between seropositivity to measles and rubella. While there are multiple explanations for the observation, and given the available evidence, the most likely is the reflection of historical and possible current community rubella transmission. Current community transmission may go relatively undetected, as 25–50% of rubella infections are subclinical or asymptomatic, especially in young children [36]. Historical transmission is supported by confirmed evidence of a congenital rubella syndrome outbreak in Samoa in 2003, resulting in 1909 reported cases (primarily in the 0–14 years age group) and three deaths [37]. Regarding age-group differences, we can infer that, in 2018 and 2019, individuals aged 14 years and under would have likely received the measles/rubella (MR) vaccine. Comparatively, seropositive individuals aged over 20 years would more likely have immunity following natural rubella infection rather than vaccination as the MR combination vaccine was introduced in 2003 and routine MMR vaccination is completed by 2 years of age, with catch-up doses recommended for children aged 1–5 years. It is also possible that age-related differences may reflect disruption to immunisation services following natural disasters. In September 2009, Samoa experienced an 8.3 magnitude earthquake and related tsunami, destroying 20 villages on the southside of Upolu and causing widespread power outages and population displacement [38]. It is possible that this led to damage to health care facilities and cold-chain systems, and prolonged interruption to routine health services, which may explain the comparatively low immunity observed in the 10–15 years age groups, who were infants or young children at the time. Alternatively, this discrepancy may also reflect differential vaccine immunogenicity, such as increased sensitivity of the measles compared to rubella vaccine to cold chain disruption [39,40], or more rapid waning of measles immunity post-vaccination [41]. These findings highlight the complexity of interpreting seroprevalence data for VPDs, since antibody responses cannot easily distinguish between vaccination and natural infection, particularly when multiple diseases are considered simultaneously. Our findings highlight the need to strengthen continued congenital rubella surveillance alongside increasing vaccination coverage, particularly in areas with historically low seroprevalence.

In the current global health climate where funding restrictions have prevented the roll-out of national surveys such as the Multiple Indicator Cluster Survey (MICS), identifying tools to estimate national VPD surveillance are crucial. The results from our study are consistent with prior estimates of vaccination coverage from previous surveys while adding additional insight into immunity levels beyond childhood and pregnant and child-rearing women. Furthermore, it is important to note that most vaccination coverage estimates, such as those produced by MICSs, are generally based on self-reported vaccination history [42] and may be influenced by factors such as desirability bias, in which people may over-report vaccination uptake. Furthermore, it is important to note that vaccination coverage estimates provide an indirect measure of population protection and rely on reporting accuracy that may overestimate population immunity (potentially due to administrative errors or inaccurate population denominators, for example), whereas seroprevalence provides more accurate estimates of true population immunity, including immunity acquired from natural infection. In this study, we identified 102 (1%) individuals seronegative to all VPDs (measles, rubella, diphtheria, and tetanus), with an average age of 11.4 years, indicating the majority were likely zero-dose children. This study highlights the utility of ISS to provide valuable information in the absence of national VPD surveys. Additionally, ISS can provide complementary information to national VPD-specific surveys to address the limitations of self-reported vaccination history.

The addition of spatial analysis further enhances the application of ISS to provide targeted information by identifying geographic patterns of immunity. Our results highlighted the notable heterogeneity in seroprevalence between PSUs, despite close geographic proximity. This pattern may be reflective of the influential role of traditional local governance structure (*faigānu’u*), which plays a central role in shaping community attitudes and behaviours [43]. The addition of spatial clustering can allow further identification of geographic patterns that may not be apparent from national or PSU-level seroprevalence estimates alone. Notably, the largest geographic cluster of seronegativity to measles includes the capital city, Apia. One possible explanation may relate to international migrants primarily settling close to the capital, where some older age groups may not have had a previous vaccination or have differing attitudes to vaccination in general.

Additionally, this cluster of seronegativity contains the Tupua Tamasese Meaole (TTM) Hospital, the major centre for national health services [44]. This may be reflective of the changes in healthcare administration regarding immunisation service delivery. In brief, immunisation services moved from community-led women’s groups (*komiti tūmamā*), formed in the 1930s under New Zealand’s administration, to more centralised health-centre based services from the 1980s to early 2000s [45,46]. In the 2010s, TTM hospital was considered the central immunisation site for Samoa; however, workforce shortages, combined with competing demands, may have impacted the availability of services [47]. Regarding access in urban and rural health centres, vaccination services are generally only available during working hours on weekdays, resulting in limited access for the working population [19]. Additionally, the rise in the anti-vaccination movement in Samoa, exacerbated by misinformation spread on social media platforms [42,48], may be more concentrated in urban areas with greater access to internet platforms [49]. Importantly, while policy reforms introduced from 2020 onwards have sought to strengthen access and delivery of vaccination in both urban and rural areas [19], the act of receiving vaccination is contingent on the intersection of individual attitudes, broader contextual influences, and the ability to physically access services. When considered together, these findings suggest that barriers to vaccination may not be entirely attributable to geographic factors.

The main limitation of this study is the lack of internationally recognised seropositivity cut-offs for MBAs and the absence of validation of tests such as sensitivity/specificity vs. gold standard, inter-assay and intra-assay variability, reproducibility, and batch-to-batch variation. Specific to VPDs, calculating seropositivity using dilution series of WHO international reference sera can enable cross-setting comparisons [50]. However, these reference values were determined to provide a clinical comparison, which may be less suitable in an epidemiological context [51]. Finally, the long time lag between data collection and the availability of MBA results was largely related to laboratory capacity and the availability of antigen-coupled beads, which are not generally commercially available and can be difficult to produce [7]. By increasing the capacity for laboratory analysis to be conducted at a regional or national level, the time needed for valuable information to be provided to the appropriate health authorities could be decreased [5]. However, ISS using MBAs currently remains primarily a research approach with important limitations outside large facilities in high-income countries.

Additional limitations to this study relate to the opportunistic use of samples collected primarily for LF. Importantly, data regarding the vaccination status of the participants would have allowed comparison of individual seropositivity to self-reported vaccination status. However, more recent SaMELFS surveys have included self-reported vaccination history in the questionnaire. Furthermore, an additional limitation is that sampling was restricted to individuals aged 5 years and older. This reflects a limitation of sampling design rather than ISS itself, as age-related differences may introduce bias or underestimation [5]. As ISS is scaled up, future surveys can address this through improved sampling design that include pre-school-aged children and considers multiple diseases or types of diseases, making this an important recommendation for planning. Convenience sampling was used because of pragmatic reasons, including cost and logistics, and was only performed for children aged 5–9. Given the large sample size in this age group, sampling bias was not considered a major issue because a very large proportion of children in each village was sampled. Despite these limitations, this study adds further knowledge on the population-level immunity to VPDs in Samoa and the utility of serosurveillance using MBA to fill information gaps when vaccination coverage data is unavailable.

In conclusion, this study builds upon previous work examining seroprevalence to infectious diseases in Samoa in 2018 and highlights the value for ISS to estimate population immunity to VPDs and reveal patterns that may be overlooked when undertaking a siloed approach. The valuable and impactful information generated by this study has led to planning for a national VPD-specific serosurvey in Samoa in 2026, which may reveal renewed declines in higher-risk areas if underlying factors contributing to low seroprevalence continue to persist. Despite limitations, the utilisation of MBAs for laboratory analysis enables a cost- and time- effective method for determining seropositivity to multiple VPDs simultaneously, which is ideal for use in low-resource settings. Combining ISS with spatial analysis enables the identification of geographic areas with higher potential for outbreak, offering an early warning system for targeted public health interventions. Moving forward, addressing key limitations such as time-lag, interpretation of results, and sampling design representativeness will be crucial to make ISS more suitable for integration into national policy.

## Figures and Tables

**Figure 1 tropicalmed-11-00009-f001:**
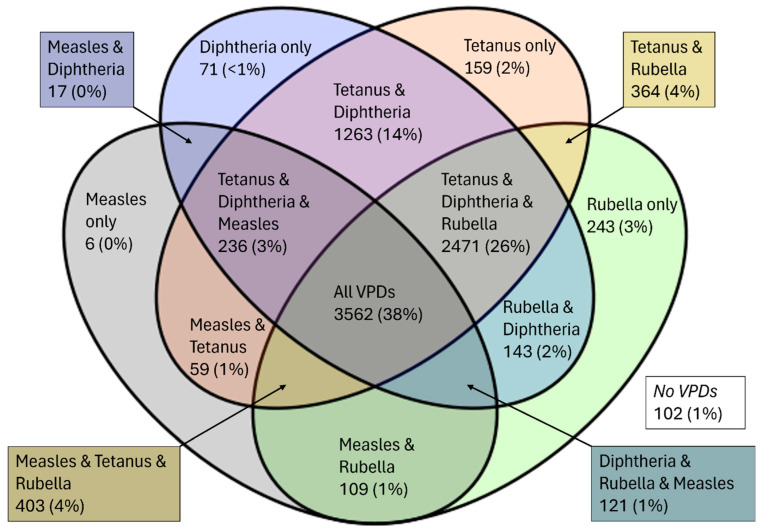
Venn diagram to show number of seropositives and seroprevalence ^a^ (n, %) to single and combinations of vaccine preventable diseases, Samoa 2018 and 2019. ^a^—Not adjusted for study design.

**Figure 2 tropicalmed-11-00009-f002:**
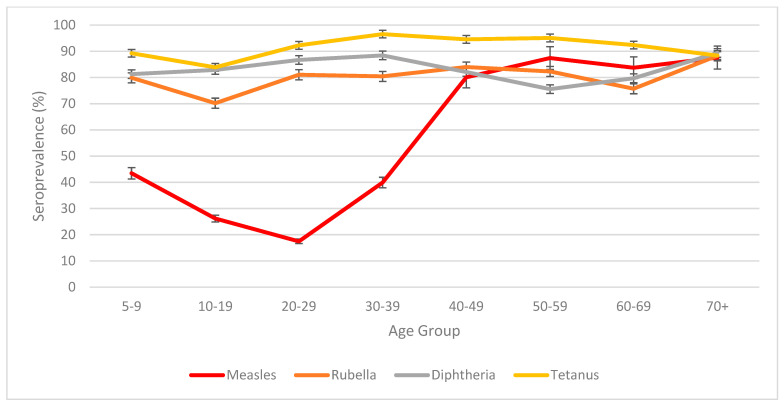
Age group seroprevalence estimates ^a^ (%) for ≤0.01 IU ^b^ protection to measles, rubella, diphtheria, and tetanus with 95% confidence intervals (CI) in Samoa 2018 and 2019. ^a^—Adjusted for study design and standardised to age and sex ^b^—International units.

**Figure 3 tropicalmed-11-00009-f003:**
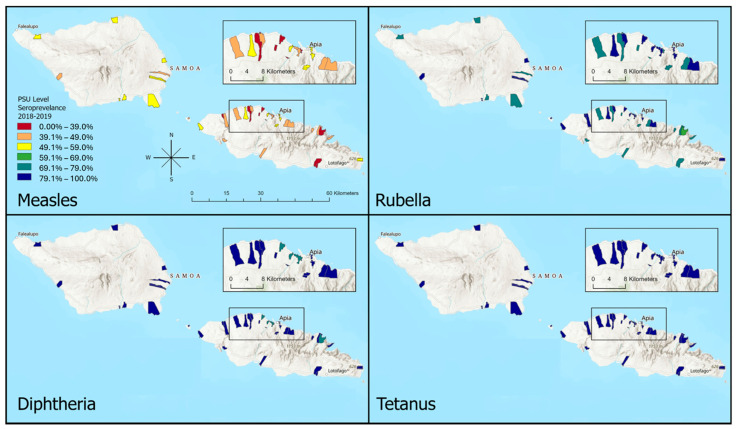
Map of primary sampling unit (PSU) level seroprevalence estimates ^a^ (%) to vaccine preventable diseases (≤0.01 international units) for measles; rubella; diphtheria; and tetanus in Samoa 2018 and 2019. This figure was created using ArcGIS Pro (Version 3.1.0). a—Adjusted for study design and standardised to age and sex.

**Figure 4 tropicalmed-11-00009-f004:**
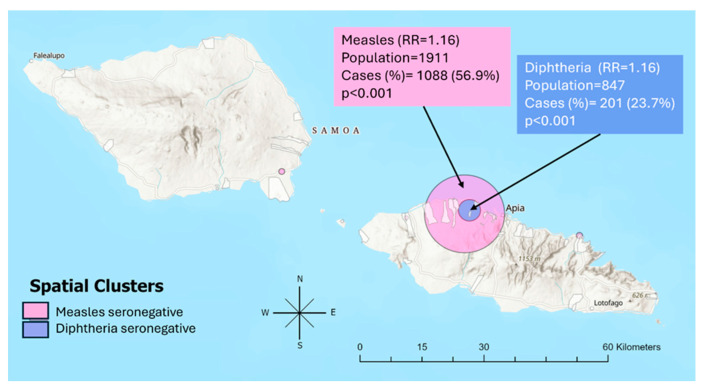
Map of spatial clusters of seronegativity for measles and diphtheria. Two statistically significant clusters (*p* < 0.05) are highlighted with smaller non-statistically significant clusters also shown. This figure was created by the authors using ArcGIS Pro (Version 3.1.0).

**Table 1 tropicalmed-11-00009-t001:** Population-level (n = 8394) seroprevalence estimates (%) by level of protection (≥0.01 IU ^a^; ≥0.1 IU; and ≥1 IU) with unadjusted and adjusted ^b^ seroprevalence with 95% confidence intervals (CI) in Samoa 2018 and 2019.

	Measles	Rubella	Diphtheria		Tetanus		
	≥0.01 IU/mL	≥0.01 IU/mL	≥0.01 IU/mL	≥0.1 IU/mL	≥0.01 IU/mL	≥0.1 IU/mL	≥1 IU/mL
Number of seropositives	4037	6664	7092	3407	7663	5942	5590
Unadjusted seroprevalence % (95% CI)	48.1% (47.3–49.3)	79.4% (78.6–80.3)	84.5% (83.7–85.2)	40.6% (39.6–41.6)	91.3% (90.7–91.8)	70.8% (69.8–71.7)	66.6% (65.6–67.5)
Adjusted seroprevalence % (95% CI)	45.8% (44.8–46.9)	79.3% (78.2–80.3)	83.7% (82.7–84.7)	31.7% (30.5–32.8)	91.0% (90.2–91.7)	70.5% (69.3–71.6)	66.0% (64.7–67.2)

^a^—International units, ^b^—Adjusted for study design and standardised to age and sex.

**Table 2 tropicalmed-11-00009-t002:** Primary sampling unit (PSU) level seroprevalence estimates ^a^ (%) by level of protection ≥0.01 IU/mL ^b^; ≥0.1 IU/mL; and ≥1 IU/mL in Samoa 2018 and 2019. Measles, rubella, diphtheria and tetanus colour-coded by percentage ranges: red (0–39.0%), orange (39.1–49.0%) yellow (49.1–59%), light green (59.1–69%), teal (69.1-79%) and blue (79.1–100%), with a separate category for No VPDs 0% (lime green).

Region	Primary Sampling Unit	Seroprevalence (%)								
		Measles	Rubella	Diphtheria		Tetanus			All VPDs	No VPDs
		≥0.01 IU/mL	≥0.01 IU/mL	≥0.1 IU/mL	≥0.01 IU/mL	≥0.01 IU/mL	≥0.1 IU/mL	≥1 IU/mL		
**Apia Urban Area**	Vaivase Tai	50.9	88.8	79.3	28.0	92.3	72.5	65.2	35.6	0.0
	Vaiala Tai + Vaiala Uta	52.9	86.4	90.6	35.4	93.3	78.9	71.8	45.1	0.0
	Avele + Letava	50.9	76.9	85.3	28.3	92.4	76.8	71.4	38.4	3.0
	Fugalei + Vaimea	31.0	76.5	73.8	25.0	88.5	63.4	58.2	22.6	1.1
	Vaimoso	47.6	82.3	69.7	18.2	83.0	47.4	46.1	19.4	2.4
	Vaitoloa	44.7	81.8	78.6	25.9	89.9	65.2	61.7	34.1	2.6
**Northwest Upolu**	Letogo	39.9	76.3	85.1	29.3	90.3	68.6	63.6	33.9	0.0
	Vaiusu	49.6	83.0	77.8	29.0	89.3	69.5	66.4	31.1	4.2
	Puipaa	36.8	85.4	68.5	27.7	89.2	68.7	66.9	25.9	1.8
	Ululoloa	35.3	72.0	89.0	37.8	87.3	68.6	65.0	20.9	2.0
	Vaitele Fou	38.4	71.2	84.4	36.1	94.5	72.8	68.7	34.4	1.0
	Lotosoa	52.1	82.5	79.2	29.0	93.0	73.6	68.6	14.2	1.6
	Nuu	38.3	78.9	85.6	30.8	92.6	72.2	67.5	25.8	0.7
	Tuanai	42.1	81.0	79.1	25.5	92.7	66.0	60.8	21.8	0.7
	Fasitoouta	45.7	83.5	87.7	41.3	94.5	67.0	64.6	18.8	3.3
	Vailuutai	37.8	85.6	85.8	30.4	92.3	76.8	71.0	33.7	3.0
	Leauvaa	37.0	74.8	85.6	26.0	90.7	64.8	59.2	27.8	5.1
	Fasitoo Tai	49.0	82.6	83.0	34.0	85.6	66.0	60.8	20.7	2.5
	Faleasiu	41.0	76.1	80.2	28.2	85.8	55.3	52.6	24.8	1.3
	Laulii	44.9	80.4	81.9	30.0	89.3	66.9	61.0	28.5	2.3
**Rest of Upolu**	Fusi	40.8	76.1	84.2	29.5	95.3	70.9	69.1	27.4	0.0
	Faleseela	46.0	71.4	84.8	36.9	88.3	66.7	58.2	31.2	1.8
	Manono Uta	54.1	88.2	89.9	29.5	89.6	64.1	61.9	43.6	0.0
	Salani + Utulaelae	36.7	73.5	83.7	36.2	91.1	73.9	69.0	21.8	1.7
	Mutiatele + Saleaaumua	49.4	80.3	88.3	34.8	97.5	75.7	70.7	24.9	0.0
	Falefa	38.0	66.5	77.2	29.9	90.0	74.5	67.6	21.0	1.9
	Faleapuna + Musumusu	41.5	77.1	88.2	29.3	93.9	78.1	71.2	42.6	0.0
	Salua (Manono Island)	43.7	86.8	86.0	33.3	90.3	72.3	68.6	25.2	0.0
**Savai’i**	Lalomalava + Safua	55.4	80.3	88.2	40.7	95.6	79.9	73.8	14.8	0.0
	Lano	50.2	84.6	94.1	38.8	95.0	82.6	71.7	21.7	0.0
	Safotu	49.3	77.1	88.6	43.1	94.0	78.4	75.6	31.4	0.8
	Sataua	49.2	73.4	90.0	36.3	91.9	75.7	69.7	32.2	0.0
	Sagone	47.5	85.2	88.4	37.7	95.5	83.3	77.5	32.0	0.0
	Papa + Tafua	49.6	76.0	83.4	27.5	86.1	60.4	57.2	26.5	1.3
	Salelolonga	51.8	76.6	84.5	25.8	88.5	73.3	70.1	16.0	1.9
**Heat scale:**		0.0–39.0% 39.1–49.0% 49.1–59.0% 59.1–69.0% 69.1–79.0% 79.1–100%							No VPDs: 0.0%	

^a^—Adjusted for study design and standardised to age and sex, ^b^—International units.

## Data Availability

The data described in this article is not available due to ethical considerations.

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
