# Peer review of "Finding the Gaps: Integrated Serosurveillance and Spatial Clustering of Vaccine Preventable Diseases in Samoa, 2018–2019"

_tropicalmed, 2025, doi:10.3390/tropicalmed11010009_

Round 1

Reviewer 1 Report

Comments and Suggestions for Authors

Review of Research Article tropicalmed-4027418

Finding the gaps: Integrated serosurveillance and spatial clustering of vaccine preventable diseases in Samoa, 2018-2019

The manuscript describes the results of two surveys which collected seroprevalence data on vaccine preventable diseases (VPDs) in Samoa. This study is important as it demonstrates the utility of integrated serosurveillance and provides a clear example of how serological data can supplement programmatic data. The manuscript is well written. There were just a few useful issues not addressed in the manuscript. Please see comments below.

Major

The finding of high rubella positivity, but considerably lower measles positivity was an interesting/confusing finding given these vaccines are given together. If your hypothesis of rubella transmission is true, do you have any programmatic recommendations from these results? Should follow-up happen in identified hot spots? Could you recommend that your spatial data could help to increase surveillance of CRS in the country?

It is still not clear as to why the measles seropositivity would drop so considerably between those in ages 40-40 and those 20-29 years. Do you have any historical records suggesting a change in measles vaccination policy over those years? Some thoughts on this would be helpful to the reader. Lastly, the authors should note in the discussion that according to their sero data, the population in Samoa (especially in the youngest) is well below the coverage targets, and the coverage reported by the Ministry as described in the Introduction. If you don’t agree with this interpretation this should be discussed as well.

The zero dose children (or individuals) is an important finding. It might have been an insightful secondary analysis if you could look to see if there is spatial clustering of zero dose individuals? Or low dose individuals, like those with 0 or 1 type of positive? To demonstrate a “worst case” type of scenario.

While not currently listed as a limitation, the authors should discuss the fact that they did not have data from children 1-5 years old. Certainly, this would have helped complete the picture, as these are vaccines given to young children, and many young children were affected by the outbreak in Samoa in 2019 (as explained in the Introduction). Related to that, the paper would be enriched by a discussion of the choice of sample in ISS generally, and how those decisions can lead to gaps for some disease-groups. Any solutions the authors can offer would be helpful to hear.

Related to the above, did the author’s hotspot identification line up with where the 2019 measles outbreak was the worst?

The authors note that a key limitation is the lack of data on vaccination status of participants. From the general design of ISS, which is often not conducted by vaccination programs specifically, it seems like this will continue to be the case. Can the authors propose any type of study or approach where seropositivity is mapped to coverage (either individually or spatially) to better understand these two methods of assessing vaccination.

Minor

Methods-please list the source of data for the PSU selection. While this may be in the previous papers it is helpful to report it here.

Line 183-184, please list the response rate with the number 8394 as is done in the earlier sentence.

Line 190-191-is “(3%)” a typo? Do you mean (38%)?

Line 211-the text refers to Table 3. This was confusing. My version of the manuscript did not have a table 3.

Table 2. I realize it already contains a lot of data, but the authors should consider a column showing the percentage of individuals seropositive to all 4. This would be interesting, and it would match the text.

Figure 4-consider using bold outlines for the dots on the map, as it is hard to see the small dots.

Line 273-276-this sentence is confusing because it says the ISS data, like that reported in this study, can fill the gap in the absence of national surveys. However, wasn’t this study a national survey? Do they mean surveys which assess single antigens only? Some other type of survey?

Reviewer 2 Report

Comments and Suggestions for Authors

Reviewer’s Comments to Authors:

  • Kindly check the punctuation marks throughout the text.
  • The authors are requested to avoid ambiguous use of the terms “minimum protection” and “seroprevalence to protection” throughout the manuscript.
  • Grammatical mistakes were observed at some places in manuscript.
  • Put through revision on typographical errors throughout the manuscript.
  1. Abstract
  • Line 22: ‘in’ Lalomalava
  • Line 24: were ‘also’ identified
  • Line 30: interventions ‘in’ ‘higher-risk’ areas
  • Abstract should avoid overstated interpretations and add limitations of this study.
  • The abstract currently gives only a limited snapshot of the study and lacks key aspects that would help in highlighting its novelty, clinical value, and broader relevance.
  1. Introduction
  • Line 38: add ‘the’ before World Health Organization
  • Line 55: use ‘low-resource’
  • Line 82: use ‘measles-containing’
  1. Materials and Methods
  • Sub-head 2.2. Data Sources: How the SaMELFS project was merged or considered to monitor the VPDs namely tetanus, diphtheria, rubella and measles is not clearly mentioned.
  • Line 125: check the sentence, mention DBS diluting media.
  • The authors are requested to explain why convenience sampling method was followed in this study? Because convenience sampling can create artifacts in the data subset and not representative of the full population.
  • The authors are requested to explain why the data from 2018 (Sep–Nov) and 2019 (Mar–May) were combined? The authors are requested to analyze the data separately for 2018 (Sep–Nov) and 2019 (Mar–May).
  • The authors are requested to add a short table in main text or supplement containing validation of test results like sensitivity/specificity vs gold standard, inter- and intra-assay variability, reproducibility, and batch-to-batch variation.
  • The authors are requested to add vaccination history.
  • The authors are requested to add the sample size.
  1. Results
  • Sub-head 3.1. Study Population Demographics: How the mean age of the participants was 19.9 years and results of ‘Seroprevalence by age and gender’ (Figure 2) showing data up to 70 years of age? Justify.
  • The authors are requested to present age distribution histogram.
  1. Discussion
  • Line 251, 258: with ‘the’ rubella vaccine
  • Line 263, 305: rephrase the sentences
  • Line 304: correct the spelling of ‘beads’
  • The authors mentioned that higher rubella seroprevalence implies community rubella transmission, which is an over-interpretation. Because higher rubella titres than measles might reflect vaccination history.
  • Discussion section needs more improvisation.
  1. Conclusions
  • Conclusion should avoid overstated interpretations and add limitations of this study.
  1. References
  • Insufficient and not written in a uniform manner, please check.

Authors are requested to check and format once again all the references according to journal format especially while abbreviating the Journal names.

Comments on the Quality of English Language

Standard English is used to write the manuscript, though rephrasing for a few sentences and correction for a few punctuations are needed

Reviewer 3 Report

Comments and Suggestions for Authors

Refer to the attached

Reviewer 4 Report

Comments and Suggestions for Authors

In the attached file.

Round 2

Reviewer 3 Report

Comments and Suggestions for Authors

All previous comments have been addressed.